# Yields, Calorific Value and Chemical Properties of Cup Plant *Silphium perfoliatum* L. Biomass, Depending on the Method of Establishing the Plantation

**Marek Bury** [1,*], **Ewa Możdżer** [2], **Teodor Kitczak** [2], **Hanna Siwek** [3] **and Małgorzata Włodarczyk** [3]

1   Department of Agroengineering, West Pomeranian University of Technology Szczecin, Pawła VI No. 3, 71-459 Szczecin, Poland
2   Department of Environment Management, West Pomeranian University of Technology Szczecin, Słowackiego 17, 71-434 Szczecin, Poland; ewa.mozdzer@zut.edu.pl (E.M.); teodor.kitczak@zut.edu.pl (T.K.)
3   Department of Bioengineering, West Pomeranian University of Technology Szczecin, Słowackiego 17, 71-434 Szczecin, Poland; hanna.siwek@zut.edu.pl (H.S.); malgorzata.wlodarczyk@zut.edu.pl (M.W.)
*   Correspondence: marek.bury@zut.edu.pl; Tel.: +48-914-496-301 or +48-606-702-560

**Abstract:** *Silphium perfoliatum* L. (*Silphium*) is one of the most promising perennial herbaceous plants, mainly due to its high biomass yield and multiple uses. It can be grown as a fodder, ornamentally, for energy (mainly as a biogas source), and as a honey crop (source of nectar and pollen for pollinators). Despite the considerable qualities of this crop, the *Silphium* cultivation area in Europe is small. The main limiting factors are the significant costs of plantation establishment and the lack of biomass yield in the first year of cultivation. Considering these aspects, research was undertaken at the Agricultural Experimental Station Lipnik of West Pomeranian University of Technology Szczecin, to assess two methods of establishing a plantation: generative, by sowing seeds (seeds); and vegetative, by transplanting seedlings grown from seeds (planting), on the yield and quality of *Silphium* biomass attended for combustion and its heating value and chemical composition. In 2016–2019, annual aboveground biomass was harvested after the end of vegetation to obtain the raw material for combustion. The collected dry mass yield (DMY) of *Silphium* significantly differed between the years and methods of establishing the plantation. The biomass yields increased in the first two years of full vegetation from 9.3 to 18.1 Mg·ha$^{-1}$·yr$^{-1}$, and then decreased in the third year of vegetation to ca. 13 Mg·ha$^{-1}$·yr$^{-1}$ because of drought. Significantly higher DMY was obtained by sowing seeds (ca. 13.9 Mg·ha$^{-1}$·yr$^{-1}$) compared to the planting method (ca. 13.0 Mg·ha$^{-1}$·yr$^{-1}$), due to the higher plant density obtained after the sowing method compared to the planting method. The calorific value in the third year was the highest and amounted to ca. 17.8 MJ·kg$^{-1}$ DM. The paper also presents changes in soil chemical properties before and after four years of *Silphium* cultivation.

**Keywords:** methods of establishing a plantation (seeds; planting); biomass yield; heating value; chemical composition of biomass; chemical properties of soil

## 1. Introduction

Poland is perceived as a country with great possibilities of biomass production for energy purposes and biofuels. This is due, according to Kuś and Faber [1], to the quite large area of arable land per one inhabitant (0.41 ha), while in the old EU countries it is two times smaller (0.19 ha). Estimates show that in Poland, between 1.0 and 4.3 million ha of agricultural land can be used for growing energy crops and biofuels [2,3]. Currently, biomass is the main source of renewable energy in Poland and in the European Union [4]. It is obtained from organic municipal waste, urban greenery, agricultural and

forestry waste and the wood industry. The supply of biomass for energy purposes can be supported by field crops of perennial plants such as: wood (*Salix spp*.), semi-wood (*Sida hermaphrodita* Rusby) and straw (*Miscanthus* sp., *Spartina pectinata*, grasses). Fuels fabricated from biomass are used in the production of heat, electricity and transport [2]. In addition, reducing fossil energy resources and the threatening of climate change are contributing to the search of new energy sources.

In Poland, the most commonly used crop for biomass production is short rotation coppice willow (*Salix spp*.). But willow requires more water and organic soils [4] or more fertile soils in which its yield is bigger [5]. The other perennial energy crops are needed to be complete or replace willow as alternatives like *Miscanthus* [6,7] and other high-yielded perennial grasses and crops. These perennial crops should be growing especially on very poor soils (marginal soils), on which agricultural crops yielded very low. Marginal soil areas in Poland are quite big and take an area of ca. 2.5 million hectares, which is 14.5% of arable land [8].

Cup plant *Silphium perfoliatum* L. is a novel high-yielding perennial crop belonging to the *Asteraceae* family and genus *Silphium*, which also includes according to Clevinger and Panero [9] many species belonging to two sections based upon root type and growth form (*Silphium* and *Composita*). To Section *Silphium* belong *S. asperrimum, S. asteriscus, S. brachiatum, S. gracile, S. integrifolium, S. mohrii, S. perfoliatum, S. radula, S. trifoliatum* and *S. wasiotense*. These taxa have fibrous root systems and a caulescent growth form [9]. *Silphium albiflorum, S. compositum, S. laciniatum*, and *S. terebinthinaceum* belonging to Section *Composita*, which have tap roots and a tendency for a scapose inflorescence with prominent basal rosettes. Clevinger and Panero [9] further wrote that the cup plant originated from North America (USA and Canada) and is native to "Great Plains Area" (*S. perfoliatum* was found in Alabama, Kansas and North Dakota).

Cup plant can be grown and harvested for over 15 years. It is originated from north-east part of North America (USA and Canada) and it found good agro-ecological conditions for growth in Europe. So far it has been used as a decorative, ornamental and honey crop (source of nectar and pollen for honey bees and other pollinators) [10,11], or as feed. Recently, *Silphium*, an energy crop is generally used for biogas production, if the biomass is harvested before the first frost like other energy crops [12–15]. As a perennial plant, it requires less finance support for tillage and less fertilizers and plant protection agents compared to annual plants [16,17]. According to McCalmont et al. [18], perennial energy plant species accumulate large amounts of organic carbon in the soil, and some crops grown for biomass can sequestrate carbon to 2.2 Mg·ha$^{-1}$ over a 20-year period.

The use of *Silphium* is not limited to anaerobic digestion, but it is a valuable raw material for the pharmaceutical and food industry. *Silphium* roots and rhizomes contain inulin [19]. *Silphium* tissue extracts have analgesic, diaphoretic, antibacterial, expectorant and cholesterol lowering properties [19,20]. The time of harvest determines the use of the Silphium biomass. If the green biomass is to be used for biogas production, it should be collected in autumn, which allows for obtaining high methane yields, accounting for more than 4000 m$^3$·ha$^{-1}$ [21,22]. However, if perennial crops are to be used for heat generation (combustion), then their harvest should take place in the period in late autumn, and even in winter or spring as *Miscanthus* [6,23]. This is justified by the increase in lignin content and decrease in moisture content, which are the result of the late development stage of the plants —growth stage 95–99 (senescence and dormancy) after Meier [24]. Is such a harvest possible for *Silphium*? What is the moisture content in cup plant biomass at harvest time in winter? Most of the conducted research concerned the use of cup plant to biogas production [15,21,22], and, also, most of the farmers in Germany are growing cup plant as a biogas source, while there are only a few publications on the use of dried biomass of cup plant for combustion [11,25,26].

The aim of the present study was to evaluate the influence of two methods of establishing the plantation: generative by sowing seeds (seeds), and vegetative by transplanting seedlings grown from seeds (planting), on the yield and quality of cup plant biomass intended for direct combustion and its calorific value. The possibility of the late harvest of *Silphium* in conditions of north-western Poland,

as well as the chemical composition and moisture content in the biomass, were examined. The changes in soil chemical properties before and after four years of *Silphium* cultivation were also investigated.

## 2. Materials and Methods

The field experiment with cup plant was carried out during four vegetative seasons (2016–2019) at the Agricultural Experimental Station (AES) Lipnik near Stargard (53°20′35.8″ N, 14°58′10.8″ E, 21 m above sea level), belonging to the West Pomeranian University of Technology Szczecin (Szczecin, North–western Poland). The cup plant in the first vegetation period after sowing or planting built only a rosette of leaves. Therefore, most of the results in this article are related to the full vegetative seasons 2017–2019.

### 2.1. Laboratory Tests

In representative soil samples taken randomly from the 0–30 cm layer in four replicates before and after the experiment, the pH, total organic matter, N, P, K, Ca, Mg, S, C and available forms of P, K, Mg were determined for each *Silphium* cultivation method from sowing and planting in the laboratories of the Department of Environment Management and the Department of Bioengineering (West Pomeranian University of Technology Szczecin, Faculty of Environment Management and Agriculture).

The chemical analyses were carried out according to standard procedures after Horwitz and Latimer [27] and relevant Polish Norms (PN). The content of nitrogen, organic carbon and sulfur was determined on a Coestech CNS elemental analyzer (Elementar Analysensysteme GmbH, Langenselbold/Hanau, Germany); available forms of phosphorus, potassium via the Egner–Riehm method [24]; available magnesium (PN-R-04024); total phosphorus (PN–98/C–04537–14); total potassium by flame photometry and total magnesium by atomic absorption spectrometry on a Perkin Elmer AAS 300 spectrometer (PerkinElmer, Inc., Waltham, MA, USA). The basic solution was obtained after the wet mineralization of soil material (PN–ISO 11466 and PN–ISO 11047), the pH value was determined potentiometrically in 1 mol dm$^{-3}$ KCl (PN–75/C–04540/05/01).

For the laboratory analysis, representative samples of dried plant material were taken from each plot (0.5 kg). The analytical moisture in the tested material was determined by the dryer-weight method (PN–80/G-04511). Dried biomass samples were ground in an analytical mill to 0.2 mm particles. Then, 1 g of such materials was determined at 105 °C in a RADWAG moisture analyzer (Radwag Balances & Scales, Radom. Poland) in three replications.

The heat of combustion of dry biomass was determined in an IKA C 2000 calorimeter (Hitachi High-Tech Corp., Tokyo, Japan) based on the isoperiobolic method (PN–81/G–04513). Then calorific value of *Silphium* biomass was calculated. The total nitrogen, phosphorus, potassium, carbon and sulfur content in *Silphium* biomass were determined in a Coestech CNS automatic analyzer (Elementar Analysensysteme GmbH, Langenselbold/Hanau, Germany) and the ash content according to standard methods after Horwitz and Latimer [27]. This study consisted of complete combustion and annealing of an analytical sample (1.0 g, crushed to 0.2 mm fraction) in a muffle furnace, heated to 815 °C. The slow burning method was used. All determinations were made in three replications.

### 2.2. Soil Characteristics and Weather Conditions

The field on which the *Silphium* experiment was established is characterized by rusty, incomplete soils, made of sand and light silty loamy sand, reaching to a depth to 30–35 cm, underlaid by light loam subsoil. The soil is postglacial, rust-brown sandy soil; classified as light soil of good rye complex; class IVb (after Polish classification soils are divided in classes: I-very good, II, IIIa, IIIb, IVa, IVb, V, VI-very poor); and a rooting profile of 0–35 cm. Soil contains about 59% sand, 28% silt and 13% loam, 1.3–1.7% organic matter, pH 5.6–6.4 in 1mol KCl.

Table 1 presents the soil nutrients content and soil characteristics of the experimental site before establishing the experiment in 2016. The data contained in Table 1 indicate that the soil used for the study was moderately acidic (pH KCL 5.90), the content of organic matter in the 0–30 cm layer

was 1.36%, and the content of macro elements was: N-0.92, P-0.45, K-0.62, Ca-0.78, Mg-0.90 and S-0.15 g·kg$^{-1}$ D.M. The soil abundance in available forms of phosphorus, potassium and magnesium was medium.

**Table 1.** Soil reaction and soil nutrient content at the experimental site.

| pH 1 moL KCl | Total Content | | | | | | | | Available Forms | | |
|---|---|---|---|---|---|---|---|---|---|---|---|
| | (g·kg$^{-1}$ DM) | | | | | | | | (mg·kg$^{-1}$ DM) | | |
| | O.M. | N | P | K | Ca | Mg | S | C | P$_2$O$_5$ | K$_2$O | MgO |
| 5.90 | 13.6 | 0.92 | 0.45 | 0.62 | 0.78 | 0.90 | 0.15 | 9.30 | 134.0 | 120.0 | 39.8 |

*2.3. Weather Conditions*

The area of AES Lipnik has a transitional climate (variable) oceanic-continental humid and temperate cold climate (according to Köppen–Geiger climate classification this area has a Dfb climate). Average annual temperature is 8.2 °C (= 46.8 °F), and annual precipitation is 536 mm (Table 2). According to the climatological station data (1981–2010), measured at Lipnik, the warmest months are usually July (with average temperature of 17.6 °C) and August (17.4 °C). The coldest month on average is January (−1.3 °C). The temperature conditions of spring and autumn are almost the same. As for annual rainfall, there are two roughly equal maximum rainfall levels in June and in July (62.0 and 67.0 mm, respectively), while the minimum rainfall is observed in the winter months, especially in February (26.0 mm).

**Table 2.** Monthly sum of precipitation (mm) and monthly average air temperature (°C) at AES Lipnik in 2016–2019 and in multi–year 1981–2010.

| | Precipitation (mm) | | | | | Temperature (°C) | | | | |
|---|---|---|---|---|---|---|---|---|---|---|
| | Year | | | | Mean [1] | Year | | | | Mean [1] |
| | 2016 | 2017 | 2018 | 2019 | | 2016 | 2017 | 2018 | 2019 | |
| I | 32.6 | 25.4 | 66.5 | 28.0 | 35 | −1.2 | −0.1 | 2.5 | −2.5 | −1.3 |
| II | 34.5 | 38.1 | 6.7 | 32.0 | 26 | 3.2 | 1.2 | 0.4 | −1.6 | −0.7 |
| III | 25.7 | 39.6 | 39.1 | 37.5 | 34 | 4.2 | 6.2 | 2.6 | 3.4 | 2.7 |
| IV | 25.7 | 38.8 | 29.1 | 2.5 | 38 | 8.9 | 8.0 | 13.0 | 8.6 | 7.2 |
| V | 43.7 | 91.8 | 38.5 | 39.5 | 52 | 16.9 | 14.7 | 16.7 | 13.4 | 12.5 |
| VI | 70.6 | 116.7 | 24.8 | 35.0 | 62 | 19.2 | 18.5 | 18.6 | 16.8 | 16.6 |
| VII | 68.7 | 180.3 | 117.2 | 22.0 | 67 | 19.4 | 18.5 | 20.1 | 18.4 | 17.6 |
| VIII | 41.2 | 76.2 | 12.9 | 136.0 | 54 | 18.2 | 19.4 | 20.5 | 17.9 | 17.4 |
| IX | 9.7 | 23.8 | 14.7 | 114.5 | 47 | 17.1 | 14.2 | 15.4 | 14.3 | 13.2 |
| X | 45.1 | 87.0 | 16.2 | 53.0 | 39 | 8.8 | 11.4 | 13.4 | 9.6 | 8.6 |
| XI | 50.5 | 67.8 | 5.9 | 47.0 | 41 | 3.9 | 6.2 | 8.4 | 4.0 | 3.8 |
| XII | 25.7 | 46.9 | 70.0 | 34.0 | 41 | 2.7 | 3.4 | 3.7 | −0.1 | 0.4 |
| Sum/Average | 473.7 | 832.4 | 441.6 | 581.0 | 536 | 10.1 | 10.1 | 11.3 | 8.5 | 8.2 |
| Sum/Average [2] | 304.7 | 614.6 | 253.4 | 402.5 | 359 | 15.5 | 15.0 | 16.8 | 14.1 | 13.3 |

[1] Mean for multi-year 1981–2010. [2] Sum/Average in Vegetation period (IV–X).

The study years 2016–2019 were very specific about the weather conditions compared to the average of the multi-year 1981–2010 (Table 2). Precipitation and air temperature, especially during the period of growth and development of cultivated plant, are of great importance. Their interaction may be the reason for shortening or extending the vegetation period, and thus delaying maturity at

too high or too low temperatures and little or high rainfall in autumn. The most rainfall during the growing season (IV–X) was recorded in 2017 (614.6 mm), and the least in 2018 (253.4 mm). The year 2018 was dry and higher rainfall occurred only in July—a rainfall event with 100 mm in one day (Table 2). Annual rainfall was 473.7 mm (2016), 832.4 mm (2017), 441.6 mm (2018), and 581.0 mm (2019). Both in the year of establishment of the plantation (2016) and in the second year of full vegetation (2018), precipitation was clearly lower compared to the average in the long term (536 mm) by about 12% and 18%, respectively, while in 2017 the rainfall was higher by about 55% of the sum from the multi-year 1981–2010. The precipitation in 2019 had almost the same value as in multi-years 1981–2010 (only by ca. 8% more), but the rainfall was not well distributed. Less precipitation was noted in the period from April to the end of August (important period for plant vegetation), except for the extreme event on 28th August (in two hours was fell 90 mm of rain). Furthermore, more than 240% of rain fell in September 2019 compared to average sum of rainfall in September of multi-years 1981–2010.

Regarding air temperature in all the years of the study, both the annual average and the average for the growing season were higher than the long-term average: during the vegetation season (IV–X) from 0.8 to 3.5 °C, and during the whole year-from 0.3 to 3.1 °C (Table 2). The year 2018, especially, was much warmer—by 3.1 °C compared to the same period in multi–years 1981-2010. It should be emphasized that the amount of precipitation this year (2018) was lower and very unevenly distributed.

## 2.4. Sowing and Planting Procedures of Silphium

A strict one-factor field experiment was established in May 2016 as a randomized block design in four replications with two treatments (sowing and planting). The experiment with cup plant was established by two methods: generative, by seeds sown directly in the soil (seeds); and vegetative, by transplanting seedlings at the stage of 3–4 leaves (planting) to the experimental field. Seeds as well as seedlings were bought from the company N.L. Chrestensen Erfurter Samen-und Pflanzenzucht GmbH (Erfurt, Germany) and were of the same origin. The seeds were homogeneous in size and of high quality with a germination rate of ca. 84%. Seeds were sown on 20 May 2016 with a Øyjord type precision seed drill in an amount of 3.0 kg per 1 ha (by seed weight), in row spacing of 45 cm to a depth of 2–3 cm and covered by the furrow closers in the form of a bent flat steel plate finished with so called dovetail. The soil was no compacted by rolling after sowing. The seedlings were transplanted in 20 June 2016 by hand in rows spacing of 45 cm to a depth of ca. 10 cm (the distance between the rows was the same like by seeds and the distance between plants inside rows was 50 cm), which provided a planting capacity of 44,000·ha$^{-1}$ (Figure 1). No irrigation was applied. The harvested area inside of one plot was 12.6 m$^2$ (total one plot area was 16 m$^2$). The side rows and one meter at all row ends were discarded to avoid border effects (the strict experiment was 160 m$^2$, and the entire experiment: 414 m$^2$).

Spring barley harvested for grain was the fore crop for test plants. After harvesting the fore crop, a traditional method of tillage was used (stubble cultivation and pre-winter plowing). In spring 2016, before sowing *Silphium*, identical mineral fertilization was applied in the amount of 100 kg·ha$^{-1}$ N, 35 kg·ha$^{-1}$ P and 110 kg·ha$^{-1}$ K, which were mixed with the rototiller used to soil preparation before sowing and planting. In subsequent years (in 2017, 2018 and 2019), after starting vegetation, before of re-growth of the plants, the same mineral fertilization (100 kg·ha$^{-1}$ N, 35 kg·ha$^{-1}$ P, 110 kg·ha$^{-1}$ K) was applied.

After the emergence of plants in 2016, weeds were controlled manually (per hand with hoes). There was no chemical weed control applied. The mechanical weed control was conducted only in the first year of the research. In the remaining years, the weed infestation was negligible and there was no need for its limitation. There was also no need to use pesticides, since no relevant pests or diseases have been recorded for cup plant. No other specific maintenance operations were needed.

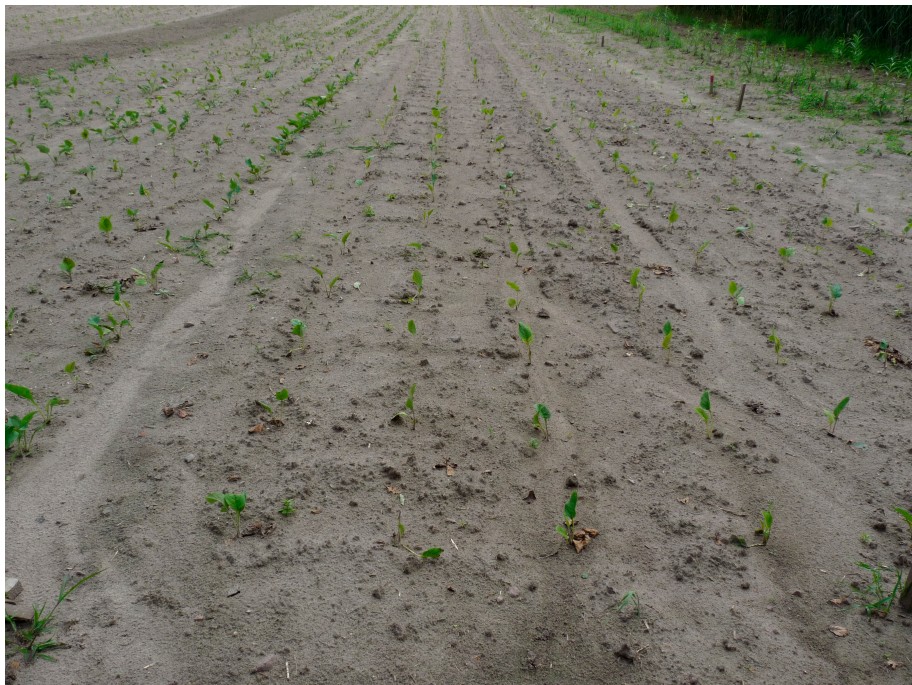

**Figure 1.** *Silphium* plants two weeks after transplanting seedlings to the field (planting).

In autumn (end of September), relative chlorophyll content as Leaf Greenness Index (LGI) was measured with the SPAD 502Plus portable Chlorophyll Meter (Konica Minolta Optics, Inc., Osaka, Japan) and the leaves assimilation area on the plant canopy as Leaf Area Index (LAI) was determined with the Ceptometer (LP-80, Decagon device, Meter Group, Inc, Pullman, WA, USA). On each plot, eight pseudo-replications (32 measurements) were performed to cover the whole plot.

## 2.5. Harvest Management

Plants were harvested from each plot separately to the end of vegetation (January 2018, 2019 and 2020) by hand. Before harvesting, the height of plants, stem (referred as a stalk or shoot), and diameter at the height of mowing (a cutting height of about 10 cm above ground) were determined using an electronic caliper (140 mm ± 0.01 mm; Limit Co.), as well as the number of shoots per plant. Plants were harvested using a petrol brushcutter, then ground in a laboratory chopper. The yields of fresh mass from each plot were weighed and the content of dry mass was determined by drying the biomass (1 kg) at 105 °C for 48 h until obtaining a constant weight (PN 8/G–04511). After that, the DMY was calculated (Mg·ha$^{-1}$).

## 2.6. Statistical Analysis

The test result was statistically processed by the ANOVA multi-way procedure using the statistical program Statistica 10.1 software (Dell Technologies, Round Rock, TX, USA). Differences between the means were assessed using the Tukey's test at a significance level of $p \leq 0.05$.

## 3. Results and Discussion

### 3.1. Plant Density

*Silphium* plants at "sowing" plots had emergences quite regularly on the surface, with a high number of plants per unit area. No problems with seed germination and emergence were observed. Schäfer et al. [28] in Germany, and Gansberger et al. [29] in Austria, emphasize the importance of seed quality and uniformity of seeds, because Silphium produces heterogeneous seeds with varying degrees of maturity with low germination capacity due to the extended ripening phase.

The transplanted seedlings of *Silphium* were growing in the year of plantation establishment (2016) and developed well (Figure 2). This was caused by high soil temperature and soil moisture, which favor faster seed germination and also better seedlings adoption. This is confirmed by Gansberger et al. [29], who claim that the ambient temperature positively affects seed germination and the juvenile growth of *Silphium*. They also reported that planting from seedlings is a more successful crop establishment method than sowing due to the poor seed quality of *Silphium* [29].

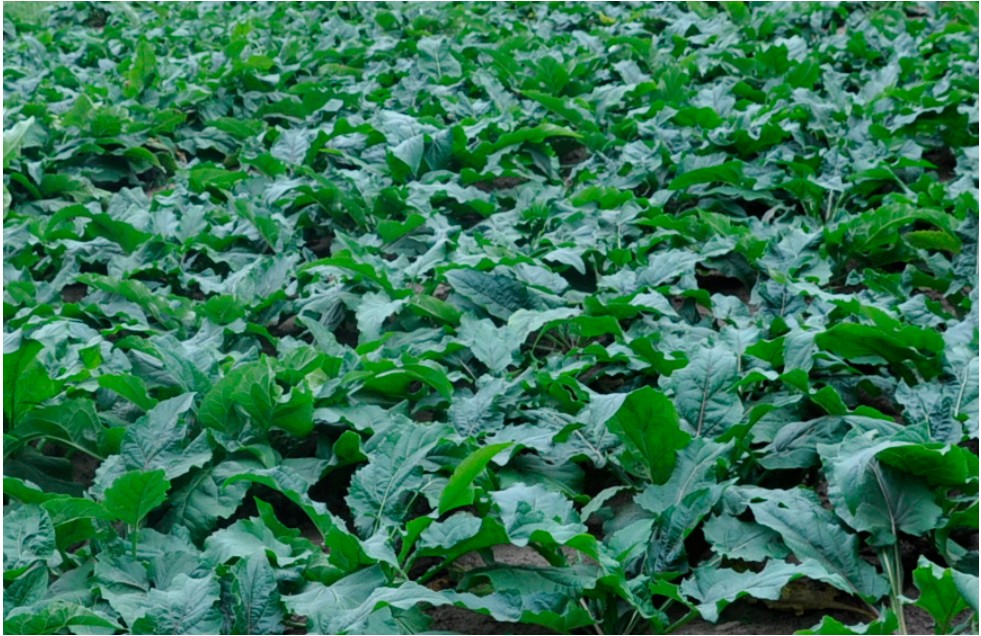

**Figure 2.** *Silphium* plants (planting) in autumn 2016—the first year of vegetation.

Sowing of 3.0 kg seeds per 1 ha (according to the recommendations of the company N.L. Chrestensen 2.5–3.0 kg) caused there to be more than three times higher plant density per unit area (1 $m^2$) in the first year compared to plots established by planting (Table 3). In the following years, there was an appreciable self-reduction in the number of plants on these plots, which decreased to ca. 10 in 2017, to ca. 7.5 in 2018 and to ca. 6 per 1 $m^2$ in 2019, which was reflected in the significance of the results. The plants in the second year of vegetation developed multi-shoot plants, which caused even stronger competition for light, water and nutrients, which intensified in subsequent years of vegetation. This was the reason for such a strong reduction in plant density in the plots established by sowing method (seeds).

**Table 3.** Plant number of *Silphium* per 1 $m^2$, depending on methods of plantation establishment.

| Year | Method of Plantation Establishment | | Mean | $HSD_{0.05}$ [1] |
| :---: | :---: | :---: | :---: | :---: |
| | **Seeds** | **Planting** | | |
| **2016** | 14.75 | 4.40 | 9.58 | 1.835 |
| 2017 | 10.30 | 4.40 | 7.35 | 2.215 |
| 2018 | 7.45 | 4.20 | 5.83 | 1.694 |
| 2019 | 6.38 | 4.20 | 5.29 | 1.346 |
| Mean | 9.72 | 4.30 | 7.01 | 0.659 |
| Mean 2017–2019 | 8.04 | 4.27 | 6.16 | 0.459 |

[1] Value means Honest significant difference at $p \leq 0.05$.

The number of plants on plots established by planting almost did not change; in the third year, the number of plants per 1 m$^2$ fell by approx. 4.5% (Table 3). In the fourth year (2019), plant density stabilization was noted, despite the fact that the plants developed many shoots. If we compare a number of plants between sowing (seeds) and planting method (planting) in full vegetation seasons (2017–2019) we found double the number of plants on sowing plots (ca. 8) compared to the planting method (ca. 4 per 1 m$^2$). This difference was significant (Table 3). Some authors [10,30] recommended four *Silphium* plants per 1 m$^2$ as an optimal density, which was achieved using the planting method, but it was a labour- and cost-intensive field establishment method. On the other hand, population density and its influence on yields were investigated by Pichard [31]. He tested different densities (from 104,000 to 208,000 plants·ha$^{-1}$) in three experimental sites, and came to the conclusion that plant density over 120,000 plants per 1ha (12 plants per 1 m$^2$) did not affect dry matter yield.

## 3.2. Plant Morphology

Plants of *Silphium* in the year of sowing (2016) developed only well-built leafy rosettes, and whole plants reached a height of approx. 30–35 cm (Figure 2). All leaves completely disintegrated during the winter.

The height of Silphium plants in full growing seasons depends on the year of vegetation and the method of establishing the plantation (Table 4). In the second year of vegetation (2017), the plants were over 2 m high. The plants on the plots established by the sowing method (seeds) were, on average, significantly smaller by about 6% compared to plants grown from seedlings. In the third year of vegetation (2018), in which the average air temperature was much higher (by 3.1 °C) and the sum of precipitation was lower—By more 94 mm than in multi-year values 1981–2010 (Table 2)—The plants were smaller by ca. 60 cm compared to the previous year (2017) and reached only 157 cm, and we did not observe a difference between plants' height depending on plantation establishing methods. In the fourth year of vegetation, Silphium plants growing on planting plots were significantly higher than plants on seeds plots (Table 4). It could be explained that many more plants per area on plots established by sowing treatment (seeds) were competing for limited resources (light, water supply and nutrients requirements). Pichard [31] obtained similar results. In his study in Chile, Pichard [31] observed an inverse linear relationship between plants height and plant density (the higher the plant density, the smaller the height of the plants) at two experimental sites (Chahuilco and Nochaco). In the third experimental site (La Unión) with favorable weather conditions, the higher population density stimulated stem elongation. Specific site conditions may have also an influence on plant growth.

**Table 4.** The height of *Silphium* plants harvested for biomass (cm), depending on methods of plantation establishment.

| Year | Method of Plantation Establishment | | Mean | HSD$_{0.05}$ [1] |
| --- | --- | --- | --- | --- |
| | Seeds | Planting | | |
| 2017 | 208 | 221 | 215 | 6.798 |
| 2018 | 156 | 157 | 157 | n.s. |
| 2019 | 183 | 215 | 199 | 2.905 |
| Mean | 182 | 198 | 190 | 10.155 |

[1] value means Honest significant difference at $p \leq 0.05$; n.s.—not significant.

Generally, the plants from plots established by the planting method had more space and built a closed canopy with massive shoots and large triangular leaves, and were higher in comparison with plants developed from seeds, which was expressed in a significant difference for the average of the full growing years. Wever et al. [32], by searching for the best *Silphium* accessions, noted an average plant height of 2.19 m in experimental field in Rheinbach near Bonn (North Rhine-Westphalia, Germany). Such plant height was measured in 2017, in a year with high precipitation (Table 2).

Schittenhelm et al. [33] reported the height of cup plant in many sites in Northern and Eastern Germany from 2.20 to 3.00 m, dependent on weather conditions and soil quality. However, in Poland, Tworkowski at al. [34] reported the average height of plants at 2.4 m.

*Silphium* produced multi-shoots plants from the second year of vegetation (Figure 3). The number of shoots per plant increased during the growing years from about 4 in 2017 to 5 in 2018 and 2019 (Table 5). The influence of the plantation establishing method (seeds versus planting) on the number of shoots per plant was found in the second and in the third growing season. Significantly more shoots (ca. 51%) were developed by plants grown from seedlings (planting) in 2017 and ca. 40% more shoots in 2018, because these plants had a larger soil surface and more light available (Table 5). In the fourth vegetation season (2019) the number of shoots has decreased by about 30% to ca. 5 shoots per plant, which may have been affected by the low rainfall and low temperature of the early vegetation period in this year (especially in April and May 2019—See Table 2). The number of shoots per plant on plots established by the generative method (seeds) increased from 3.5 in 2017 to about 4.5 in 2018 and 2019 (Table 5), which was caused by a decrease in the number of plants per unit area (1 m$^2$) from ca. 10 to 6 at the same period (Table 3). Wever et al. [32] in Rheinbach (Germany) reported much higher (two times) number of shoots per plant - between 9.86 and 12.31, depending on studied *Silphium* accessions. However, these high numbers of stalks per plant were connected with the low plant density the authors used (17,800·ha$^{-1}$). Pichard [31] also observed a reduction in the number of stems per plant in response to the higher plant population density at all three tested sites in Chile.

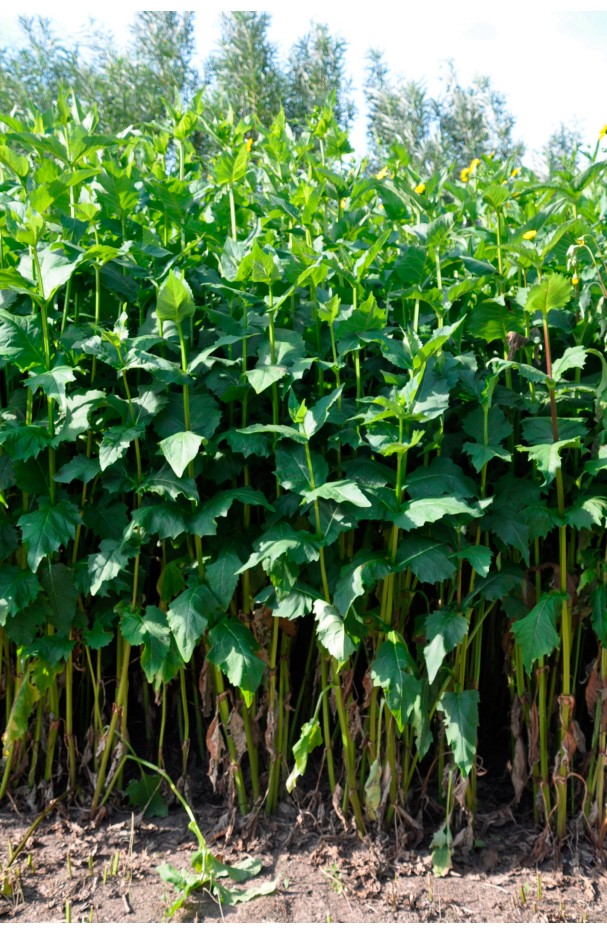

**Figure 3.** *Silphium* plants (seeds) in the second vegetation season.

**Table 5.** The number of shoots per plant of *Silphium*, depending on methods of plantation establishment.

| | Method of Plantation Establishment | | Mean | HSD$_{0.05}$ [1] |
|---|---|---|---|---|
| | **Seeds** | **Planting** | | |
| 2017 | 3.5 | 5.3 | 4.4 | 0.623 |
| 2018 | 4.5 | 6.3 | 5.4 | 1.113 |
| 2019 | 4.6 | 4.9 | 4.8 | n.s. |
| Mean | 4.2 | 5.5 | 4.9 | 0.964 |

[1] value means Honest significant difference at $p \leq 0.05$; n.s.—not significant.

The number of plants on unit area (1 m$^2$) and the number of shoots per plant affected the number of shoots per square meter (Table 6). The differences between treatments (seeds versus planting) were quite large. On the plots established by planting methods, in 2017, we found ca. 23 shoots and for seeds treatment these number of stalks per unit area were significantly higher by approximately 56%. The same relation we found also in the third (2018) and fourth (2019) year of cultivation—The number of shoots were significant higher by ca. 27 and 43%, respectively, on the plots established by the seeds method in comparison to the number of shoots on plots established by the planting method. When we compared the number of shoots per unit area (1 m$^2$), we could observe that for the higher plant density on seed treatment, the number of stalks decreased every year, and for lower plant densities on plots with the planting method, the number of shoots per 1 m$^2$ increased from ca. 23 at 2017 to ca. 27 in 2018 and then decreased to ca. 21 stalks per 1 m$^2$ in 2019 (Table 6). This decrease could be explained by drought in 2019 and the self-regulation of the canopy. The plant density in 2019 allowed the optimal use of the free space between the shoots, as well as solar radiation, especially photosynthetic active radiation (PAR).

**Table 6.** The number of shoots per 1 m$^2$ of *Silphium*, depending on methods of plantation establishment.

| | Method of Plantation Establishment | | Mean | HSD$_{0.05}$ [1] |
|---|---|---|---|---|
| | **Seeds** | **Planting** | | |
| 2017 | 36.0 | 23.3 | 29.7 | 0.291 |
| 2018 | 33.5 | 26.5 | 30.0 | 0.650 |
| 2019 | 29.4 | 20.6 | 25.0 | 1.116 |
| Mean | 33.0 | 23.5 | 28.2 | 2.897 |

[1] value means Honest significant difference at $p \leq 0.05$.

*Silphium* plants build cross-sectional characteristic shoots in the form of a rectangle. Analysis of the diameter of shoots, developed in years of full vegetation (2017–2019) showed that both the method of plantation establishment and year have influence on this trait (Table 7). In the second year of vegetation (2017), the diameter of shoots of plants grown from seedlings (planting) was significantly larger by approx. 11% than those grown from seeds. This relationship was found in the third and fourth year of vegetation, but it was not significant. When assessing the diameter of the shoots in all years of vegetation, it should be noted that significantly thicker shoots developed plants from plots established by planting, because these plants developed higher shoots (Table 4) and diameter of stem is correlated to the height—the higher the plants, the thicker they are. Wever et al. [32], by measuring of different *Silphium* accessions, noted an average shoot diameter (at 10 cm above ground) of 18.93 mm in an experimental field in Rheinbach (Germany). It was much higher in comparison to shoot diameter in our own study. However, Frączek et al. [25], during the test of the usefulness of *Silphium* to briquetting, obtained the diameter of the shoot as the length of the side of the rectangle measuring 11.2 × 12.4 mm. These results are similar to those obtained in our own research.

**Table 7.** The diameter of *Silphium* shoots (mm), depending on methods of plantation establishment.

| Year | Method of Plantation Establishment | | Mean | HSD$_{0.05}$ [1] |
| --- | --- | --- | --- | --- |
| | Seeds | Planting | | |
| 2017 | 13.2 | 14.6 | 13.9 | 0.863 |
| 2018 | 11.5 | 11.7 | 11.6 | n.s. |
| 2019 | 12.3 | 12.9 | 12.6 | n.s. |
| Mean | 12.3 | 13.1 | 12.7 | 0.450 |

[1] Value means Honest significant difference at $p \leq 0.05$; n.s.—not significant.

In general, a single Silphium plant established by the planting method was, on average, 16 cm higher, had a thicker stem by 0.8 mm, developed more shoots per plant (by 1.3) compared to the plant established by sowing method (seeds). This resulted mainly from the plants density per unit area, because the number of plants on plots established by the planting method (planting) was approximately two times lower than the number of plants on plots established by the sowing method. These results could suggest that competition for soil nutrients was more important than light limitation with increasing plant density. Pichard [31] gives similar justification to his investigations to the impact of plant density on the morphological traits and *Silphium* yield in Chile. Schäfer et al. [28] and Gansberger et al. [29], in their research in Germany and Austria, explained that the establishment of plantation by transferring seedlings to the field causes the earlier and more regular development of plants grown from seedlings compared to plants grown from seeds. The earlier and better development of plants grown under optimal conditions in a nursery (greenhouse) could have an important influence on better rooting system development, and thus better nutrients and water uptake from deeper layers of soil.

The LAI index allows for determining the degree of light use by plants. Higher index values indicate a larger assimilation area per soil surface (1 m$^2$) and greater use of light energy in the photosynthesis process [35]. The LAI index was determined not by invasive methods, using a portable device which allows 80 different sensors to measure the photosynthetically active radiation (PAR) in the electromagnetic spectrum of 400–700 nm above canopy and close to the soil surface, and is not equal to the surface of all assimilation organs on the plant. In the conducted research, both the method of establishing the plantation and year of vegetation significantly influenced its values (Table 8). The plants from plots established by seeds were characterized by a significantly lower LAI index than those planted from seedlings (planting) in all the years of vegetation. This difference was mainly due to the number of green leaves per unit area. In the time of measurements (the end of September), the *Silphium* plants were finishing vegetation. Some of the lower leaves had fallen off completely, and some of the leaves were dried, brown-black colored, especially in the middle part of stem (Figure 4). Only the upper leaves were green and assimilated active on plots with a large number of shoots (seeds treatment). In contrast, plants on plots established by the planting method had more green leaves with larger leaf blades, which meant that their assimilation surface expressed as LAI was significantly larger, despite the smaller number of shoots per unit area.

**Table 8.** The Leaf Area Index of *Silphium* (m$^2 \cdot$m$^{-2}$), depending on methods of plantation establishment.

| | Method of Plantation Establishment | | Mean | HSD$_{0.05}$ [1] |
| --- | --- | --- | --- | --- |
| | Seeds | Planting | | |
| 2017 | 4.68 | 4.89 | 4.79 | 0.186 |
| 2018 | 4.88 | 5.29 | 5.09 | 0.160 |
| 2019 | 5.03 | 5.20 | 5.12 | n.s. |
| Mean | 4.86 | 5.13 | 5.00 | 0.076 |

[1] Value means Honest significant difference at $p \leq 0.05$; n.s.—not significant.

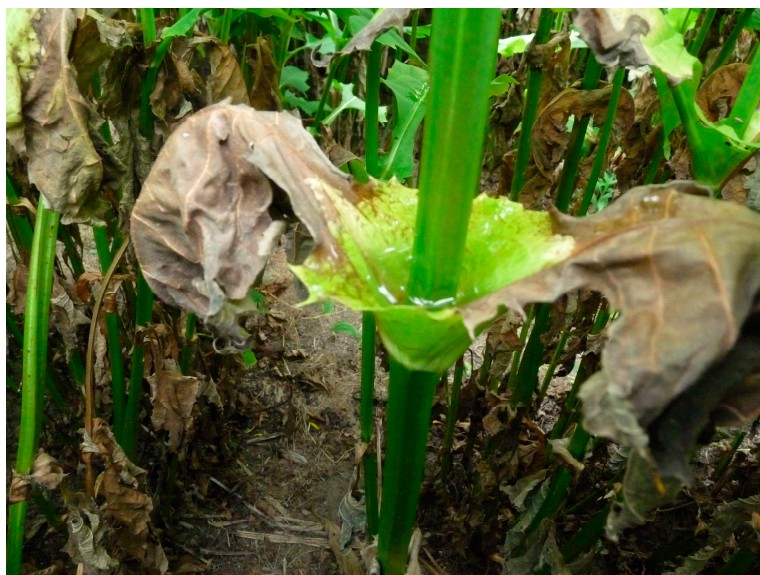

**Figure 4.** Inside of rows of *Silphium* plots (seeds) in the third vegetation season.

The chlorophyll content in the leaves could be determined indirectly by using noninvasive methods such as LGI, which show the relative chlorophyll content in SPAD units. The analysis of the LGI values of the *Silphium* plants harvested for biomass was not unidirectional depending on the method of plantation establishment and year (Table 9). In the second (2017) and fourth (2019) years of vegetation, the greenness index of plants on plots established by the planting method (planting) was higher (but not significant in 2019) compared to those LGI of plants on plots established by the generative method (seeds). In the third year of vegetation (2018), the plants on plots established by the seeds method had a higher greenness index, though only by 2%, compared to the LGI of the plants on plots established by planting method (Table 9).

**Table 9.** The Leaf Greenness Index (LGI) of *Silphium* plants (SPAD), depending on methods of plantation establishment.

| Year | Method of Plantation Establishment | | Mean | $HSD_{0.05}$ [1] |
|---|---|---|---|---|
| | Seeds | Planting | | |
| 2017 | 30.20 | 32.77 | 31.49 | 0.877 |
| 2018 | 32.70 | 32.10 | 32.40 | n.s. |
| 2019 | 34.64 | 35.81 | 35,22 | n.s. |
| Mean | 33.56 | 33.04 | 33.80 | n.s. |

[1] value means Honest significant difference at $p \le 0.05$; n.s.—not significant.

### 3.3. Dry Mass Yield (DMY)

In the year of establishing the field experiment (2016) *Silphium* developed only a leaf rosette consisting of 8–20 leaves. The leaves died completely and decomposed during the winter. Under production conditions, an aboveground mass will be not harvested, because of small yield and low economic profitability. In the years of full vegetation (from the second to the fourth growing year), cup plant was harvested in the winter between 20 and 30 January—The stems were withered with the upper leaves remaining on the plant, and the dry matter content was between 76–82%.

In the first year of full vegetation (2017), cup plant found favorable weather conditions for growth with a high average temperature and high and well-distributed precipitation (Table 2), but it did not have enough of a developed root system, because it was in a so-called juvenile stadium. The biomass yields were on average 9.32 Mg·ha$^{-1}$ and year regardless of treatments (Table 10). In the second full

growing period (2018), *Silphium* developed well by high temperature and sunlight despite smaller rainfall (ca. 82% of multi-year precipitation) and gave surprisingly high biomass DMYs, which reached ca. 18.1 $Mg \cdot ha^{-1} \cdot yr^{-1}$. DMYs were almost twice as high as compared to yields achieved in 2017. In the third full vegetation season (2019) the DMYs decreased to ca. 13 $Mg \cdot ha^{-1} \cdot yr^{-1}$. This can be explained by drought, especially in the first important part of vegetation, from April to the end of June. In this multi-year period, precipitation fell by only 45% and also August was dry, except the 28 of August with an extreme event with 90 mm of rain.

**Table 10.** DMY of Silphium ($Mg \cdot ha^{-1}$) depending on methods of plantation establishment.

| | Method of Plantation Establishment | | Mean | HSD$_{0.05}$ [1] |
| --- | --- | --- | --- | --- |
| | **Seeds** | **Planting** | | |
| 2017 | 9.77 | 8.86 | 9.32 | 0.247 |
| 2018 | 18.73 | 17.45 | 18.09 | 0.201 |
| 2019 | 13.15 | 12.78 | 12.96 | n.s. |
| Mean | 13.88 | 13.03 | 13.46 | 0.378 |

[1] value means Honest significant difference at $p \leq 0.05$.

The biomass yield of cup plant depended on plantation establishment method. The bigger biomass yields were obtained by seed treatment method (seeds). The average DMY of *Silphium* biomass from the entire study period was ca. 13.9 $Mg \cdot ha^{-1} \cdot yr^{-1}$ in contrast to DMY achieved by planting method (ca. 13.0 $Mg \cdot ha^{-1}$ and year). In the first (2017) and second full growing season (2018) the average DMYs of cup plant were significantly higher by 10% and 7%, respectively, compared to DMYs from plots established by vegetative method (planting). Moreover, DMY was also higher by seeds treatment in the third vegetation season (2019), but the difference was not significant (Table 10). The DMY of seed-treatments was higher despite the smaller height and diameter of individual shoots, and depended mainly on the greater number shoots per unit area (Table 6). Plants growing in low densities on plots established by planting, which developed higher and thicker stems, were not able to compete with a significantly greater number of plants and shoots on plots established by the generative method (seeds). Pichard [31], in his research on the effect of population density on DMY and plant morphology, confirms that the number of stems, which is the major component of the yield structure, determines the yield.

However, it should be noted that *Silphium* harvest for biomass after winter in north-west Poland is difficult from a technical point of view, because plants (dried shoots) lodged and partially lied on soil surface (it happened in January 2018 and 2019), which hindered mechanical harvesting (only harvest at 2020 was without any problem).

A similar level of yield on light soil was obtained by Schittenhelm et al. [33]. They inform the dry matter yield was in average across two years (2013–2014) in Braunschweig 10.8 $Mg \cdot ha^{-1}$ and year without irrigation and by density of 4 plants per 1 $m^2$. Furthermore, Stolarski [10] reported that the yield of *Silphium* in the conducted research in North-East Poland ranged from 11.2 to 13.9 $Mg \cdot ha^{-1} \cdot yr^{-1}$ DM. Wever et al. [32], in Rheinbach (North Rhine-Westphalia, Germany), reported yields between 8.4 to 14.3 $Mg \cdot ha^{-1} \cdot yr^{-1}$ DM, harvested in December 2016 depending on studied *Silphium* accessions, but in the same time they noted higher yields in second site, in Dornburg, between 13.59 and 17.21 $Mg \cdot ha^{-1}$ DM and year (mean 15.46 $Mg \cdot ha^{-1} \cdot yr^{-1}$ DM). [34,36–38] reported the yield from 14.0 to 19.0 $Mg \cdot ha^{-1} \cdot yr^{-1}$ DM obtained in Poland, which is close to the value of given in European literature, which was confirmed by our own research. In contrast, Stanford [39] reported very high yield by approximately 40 $Mg \cdot ha^{-1} \cdot yr^{-1}$ DM in North America. Research from Kuś and Matyka [40] showed that it is profitable to run energy crop plantations when biomass yields exceed 11.0 $Mg \cdot ha^{-1} \cdot yr^{-1}$. The analysis of our own research results could indicate that profitability of *Silphium* cultivation in both methods of plantation establishment, from sowing and planting, was achieved in the second and in the

third year of full vegetation. In addition, Kuś and Matyka [40] reported that crop yield was generally influenced by plant density.

### 3.4. Calorific Value of Silphium

The data regarding the calorific value of *Silphium* expressed in MJ·kg$^{-1}$ indicate differences both between the method of establishing the plantation and years of research (Table 11). In the first full year of vegetation, the calorific value of above ground mass was between ca. 14.6 (planting) and 15.9 MJ·kg$^{-1}$ (seeds); in the third year of the study (2018), the calorific value of the test biomass ranged from 16.5 (planting) to 17.1 MJ·kg$^{-1}$ (seeds). The biomass obtained from plants grow on plots established by seeds had significantly higher calorific value. Moreover, in the third year of full vegetation (2019), calorific value was higher on plots established by seeds-ca. 17.9 MJ·kg$^{-1}$, in contrast to plots established by planting (ca. 17.7), but the difference was not significant. These values were the highest from all investigation years. The research results indicate that the calorific value of the test biomass increased between the second, third and fourth year of cultivation. The calorific value was higher by 4.2% from plants established by sowing method (seeds). It could be explained that *Silphium* biomass at the time of harvest in winter consists almost entirely of stems, whose structure and chemical composition may vary depending on their thickness. Thin stems from plots established by the sowing method may have a more favorable chemical composition (lower moisture, higher content of cellulose, lignin, hemicellulose and others) and elemental composition (C, H, O, N) compared to thicker stems from plots established by the planting method. In our study, we found less moisture and higher C content (not significant) in biomass samples from plants established by the generative propagation method (seeds).

**Table 11.** The calorific value of *Silphium* biomass (MJ kg$^{-1}$), depending on methods of plantation establishment.

| | Method of Plantation Establishment | | Mean | HSD$_{0.05}$ [1] |
|---|---|---|---|---|
| | Seeds | Planting | | |
| 2017 | 15.89 | 14.59 | 15.24 | 0.316 |
| 2018 | 17.10 | 16.52 | 16.81 | 0.331 |
| 2019 | 17.87 | 17.68 | 17.78 | n.s. |
| Mean | 16.95 | 16.26 | 16.61 | 0.340 |

[1] value means Honest significant difference at $p \leq 0.05$; n.s.—not significant.

In the conducted tests, the calorific value of *Silphium* biomass from sowing and planting was 16.61 MJ·kg$^{-1}$ on average over three years. Similar results were obtained by other authors [10,34,37], which confirms our tests. Stolarski [10] reported that the calorific value of *Silphium* biomass was in the same range as other perennial energy crops such as *Miscanthus sinensis, M. sacchariflorus, M. x giganteus, Reynoutria sachalinensis, R. japonica, Spartina pectinata, Sida hermaphrodita* Rusby, *Rosa multiflora* and willow (*Salix spp.*). It was also found that *Silphium* biomass had good physical properties from an energy point of view, which, in combination with low water and soil requirements, makes it a valuable energy plant, thus increasing the range of crops grown for energy purposes [38].

### 3.5. Moisture and Content of Ash and Macronutrients in Silphium Biomass

The obtained test results from chemical analysis are summarized in Table 12. *Silphium* biomass moisture ranged from 3.84% to 5.46%. It was significantly diversified, both in terms of the method the plantation was established and the vegetation period. *Silphium* biomass contained the highest moisture in the first year of full vegetation (2017). The average moisture in 2017 was 4.65%. In the following years, the moisture content decreased to ca. 4.4%. It was found that the moisture content of *Silphium*

biomass was significantly higher in cultivation from planting compared to cultivation from sowing (seeds) by ca. 29%.

**Table 12.** The content of analytic moisture and ash (%) and the total content of nitrogen, phosphorus, potassium, sulfur and carbon (g·kg$^{-1}$ DM) in the biomass of *Silphium*, depending on methods of plantation establishment.

| | Method of Plantation Establishment | Year | | | Mean |
|---|---|---|---|---|---|
| | | 2017 | 2018 | 2019 | |
| Moisture (%) | seeds | 3.84 | 3.95 | 3.98 | 3.92 |
| | planting | 5.46 | 4.79 | 4.89 | 5.05 |
| | Mean | 4.65 | 4.37 | 4.44 | 4.49 |
| | HSD$_{0.05}$ [1] | 0.046 | 0.057 | 0.059 | 0.026 |
| Ash [%] | seeds | 3.26 | 4.46 | 4.53 | 4.08 |
| | planting | 2.84 | 3.22 | 3.24 | 3.10 |
| | Mean | 3.05 | 3.84 | 3.88 | 3.59 |
| | HSD$_{0.05}$ | 0.027 | 0.030 | 0.031 | 0.011 |
| N (g·kg$^{-1}$DM) | seeds | 14.55 | 17.47 | 16.68 | 16.23 |
| | planting | 13.70 | 13.66 | 13.82 | 13.73 |
| | Mean | 14.13 | 15.57 | 15.25 | 14.98 |
| | HSD$_{0.05}$ | 0.218 | 0.245 | 0.264 | 0.172 |
| P (g·kg$^{-1}$DM) | seeds | 0.47 | 0.93 | 0.89 | 0.76 |
| | planting | 0.40 | 0.71 | 0.73 | 0.61 |
| | Mean | 0.44 | 0.82 | 0.81 | 0.69 |
| | HSD$_{0.05}$ | 0.033 | 0.044 | 0.038 | 0.018 |
| K (g·kg$^{-1}$DM) | seeds | 0.68 | 1.02 | 1.06 | 0.92 |
| | planting | 0.64 | 0.86 | 0.92 | 0.81 |
| | Mean | 0.66 | 0.94 | 0.99 | 0.86 |
| | HSD$_{0.05}$ | n.s. | 0.037 | 0.031 | 0.042 |
| S (g·kg$^{-1}$DM) | seeds | 0.71 | 0.71 | 0.73 | 0.72 |
| | planting | 0.59 | 0.49 | 0.51 | 0.53 |
| | Mean | 0.65 | 0.60 | 0.62 | 0.62 |
| | HSD$_{0.05}$ | 0.040 | 0.045 | 0.039 | 0.054 |
| C (g·kg$^{-1}$DM) | seeds | 343 | 343 | 345 | 344 |
| | planting | 337 | 338 | 341 | 339 |
| | Mean | 340 | 340 | 343 | 341 |
| | HSD$_{0.05}$ | n.s. | n.s. | n.s. | n.s. |

[1] value means Honest significant difference at $p \leq 0.05$; n.s.—not significant.

The ash content in *Silphium* biomass ranged from 2.84% to 4.46%. It was significantly diversified both in terms of the method of plantation establishment and the year of vegetation (Table 12). The most ash contained *Silphium* biomass in the third year of full cultivation, on average 3.88%, which was higher compared to 2017 and 2018 by 0.83 and 0.04 units (percent points), respectively. It was found that the content of ash in *Silphium* biomass was significantly lower in cultivation from planting in relation to cultivation from sowing (seeds) by an average of 24%. Stolarski [10], in research conducted with various energy crops, obtained higher ash content in *Silphium* biomass by ca. 7.0%. Furthermore,

Wever et al. [32] noted higher ash content in *Silphium* biomass (on average 9.22%). Frączek et al. [25] reported that the *Silphium* biomass has a higher moisture content, which was 13%, while the ash content was at a similar level (3.4%).

Excessive content or deficiency of macronutrients in arable crops may reduce their quality when considered for industrial processing. It also indicates the dynamics of the transition of nutrients from soil to plants [41].

The content of nitrogen in plant biomass was significantly different. It is probably related to weather conditions, which varied widely in 2017 (high rainfall), 2018, and 2019 (drought). *Silphium* biomass contained less nitrogen in the second year of experiment (2017) in both methods of plantation establishment. It was lower compared to 2018 and 2019 by ca. 10% and ca. 7%, respectively (Table 12). In subsequent years, the nitrogen content increased, and the highest value was found in 2018. A significant difference in nitrogen content was found between methods of plantation establishment (seeds, planting)—Nitrogen content in plants biomass from sowing was significantly higher than in plants biomass from planting.

The phosphorus content of *Silphium* biomass varied significantly in the vegetation years and between methods of plantation establishment. The content of P ranged from 0.40 to 0.93 g·kg$^{-1}$ DM. These values were smaller than those obtained from Kuś and Matyka [38]. *Silphium* accumulated in plant biomass more phosphorus in the sowing establishment methods (seeds) by ca. 24% compared to plant biomass from planting method (Table 12).

The *Silphium* biomass in the first full year of vegetation contained the least potassium in plant biomass of both sowing and planting methods of plantation establishment (ca. 0.66 g·kg$^{-1}$ DM). In subsequent years, the potassium content increased and the highest value was found in 2019. These values were higher in 2018 and 2019 by ca. 42% and 50%, respectively, compared to the year 2017 (Table 12). The potassium content of *Silphium* biomass varied significantly, not only in the vegetation years, but also between methods of plantation establishment. *Silphium* accumulated more potassium in the plant biomass from sowing establishment method (seeds) by ca. 14% compared to plant biomass from planting method (Table 12).

Higher potassium content in *Silphium* biomass was noted by Łabętowicz et al. [42], who reported that potassium content amounted to 2.1 g·kg$^{-1}$ DM. Based on the data, it can be stated that in our own experiment, the content of potassium in the analyzed plant biomass was lower in both the sowing and planting method of plantation establishment. The trend of increasing phosphorus and potassium in the biomass of the test plant was found in both methods of plantation establishment in 2018 and 2019. This relationship was most likely caused by the weather conditions prevailing during the vegetation period of this crop.

The *Silphium* biomass all years of the experiment contained almost the same sulfur content -ca. 0.62 g·kg$^{-1}$ DM (Table 12). *Silphium* accumulated significantly more sulfur in the biomass from the sowing establishment method (seeds) by ca. 36% compared to the planting method. Analyzing the second and third year of research, the sulfur content stabilized and ranged from 0.60 to 0.62 g·kg$^{-1}$ DM on average. The similar sulfur content in *Silphium* biomass was achieved by Stolarski [10], who reported that it amounted to 0.69 g·kg$^{-1}$ DM.

It was also found that *Silphium* biomass has good chemical properties from an energy point of view, which in combination with low moisture and ash content, and also low content of N, P, K, and S in the biomass makes it a valuable energy plant. Thus, the range of crops grown for energy purposes should be increased [38].

The total carbon content in *Silphium* biomass was not significantly differentiated and ranged from 337 to 345 g·kg$^{-1}$ DM (Table 12). These values were smaller than those obtained by Stolarski [10]. *Silphium* accumulated more carbon (but not significant) in the plant biomass from plots established by seeds method compared to plant biomass from planting method by ca. 1.5%.

Crops that leave biomass in semi-woody forms, such as *Silphium,* with later harvesting time, are characterized by more favorable energy parameters. During favorable weather conditions by

harvest, there is a decrease in biomass moisture and an increase in calorific value. However, in conditions of worsening weather (high humidity) by the harvest, the biomass moisture increases and its calorific value - decreases. The time of harvest and the prevailing weather conditions during cultivation and harvesting have an important role in the quality of the *Silphium* biomass.

### 3.6. Soil Characteristics after Silphium Harvest

The soil pH after the experiment and harvest of the test crop in the 0–30 cm layer ranged from 5.95 to 6.05 (Table 13). Analyzing the data, it was found that the pH of the soil increased during the cultivation of *Silphium*, compared to the value recorded before of the experiment establishment (5.90) by 0.05 (sowing) to 0.15 unit (planting), which has a positive effect on the properties of the soil. Similar research results were obtained by Baran et al. [43].

**Table 13.** The characteristic of the soil taken from 0–30 layer after harvest of *Silphium* in 2019, depending on the methods of plantation establishment.

| | Method of Plantation Establishment | | | |
|---|---|---|---|---|
| | **Seeds** | **Difference** [1] | **Planting** | **Difference** [1] |
| $pH_{KCl}$ | 5.95 | +0.05 | 6.05 | +0.15 |
| **Total Content (g·kg$^{-1}$ DM)** | | | | |
| N | 0.84 | −0.08 | 0.82 | −0.10 |
| P | 0.38 | −0.07 | 0.42 | −0.03 |
| K | 0.58 | −0.04 | 0.62 | −0.00 |
| Mg | 0.30 | −0.60 | 0.34 | −0.56 |
| Ca | 0.58 | −0.20 | 0.64 | −0.14 |
| S | 0.040 | −0.110 | 0.039 | −0.111 |
| C | 7.40 | −1.90 | 7.23 | −2.07 |
| **Available Forms (mg·kg$^{-1}$ DM)** | | | | |
| $P_2O_5$ | 164.5 | +30.5 | 170.7 | +36.7 |
| $K_2O$ | 125.4 | +5.4 | 124.8 | +4.8 |
| MgO | 30.1 | −9.7 | 30.8 | −9.0 |

[1] Difference in content (after harvest/before plantation establishment).

The concentration of macronutrients in the soil after the end of the experiment (Table 13) was varied and was generally smaller compared to the content before the experiment establishment (Table 1). The content of nitrogen in the soil after the experiment ranged from 0.82 to 0.84 g·kg$^{-1}$ DM. These values were lower by 10% on average than before the experiment.

The soil after the fourth year of experiment (2020) also contained clearly less of all macro elements tested (total forms) in both plantation establishment methods; especially large losses were found for magnesium, calcium, sulfur and carbon (Table 13). Average losses of the above macronutrients, regardless of the method of establishing the plantation, were ca. 65; 22; 73; and 21%, respectively. Soil from sowed *Silphium* plots (seeds) contained less P, K, Mg and Ca than soil from plots where *Silphium* were planted (planting).

Analyzing the content of available phosphorus and potassium forms in the soil after harvesting the test crop, an increase in the content of both macronutrients was found in both the methods of plantation establishment: content of $P_2O_5$ by 22.8% and $K_2O$ by 4.5% under sowing plots (seeds) and $P_2O_5$ by 27.4% and $K_2O$ by 4.0% under planted plots (planting), compared to the value before the establishment of experiment. However, the content of available magnesium (MgO) in both methods of plantation establishment decreased by more than 20%, which is associated with low content of this

element in soil and increased uptake by plants. We think the increase in exchangeable or available forms of phosphorus and potassium in the soil is caused by leaves fall in autumn and leftover plant residues after harvest and their mineralization. It can be caused also by well-developed rooting system that takes both macronutrients from the soil and accumulates them in the topsoil. Similar results were obtained by other authors [44].

## 4. Conclusions

The conducted study showed that cup plant *Silphium perfoliatum* L. can be considered as an alternative energy crop, used not only for biogas production like nowadays, but also as a good source for the direct combustion process. The biomass yields were varied in the study years, which was related to meteorological conditions and soil fertility. The highest dry matter yield (DMY) of *Silphium* biomass on light (marginal) soil was obtained in the third year of the vegetation (the second year of full vegetation season). It was reached ca. 18.1 Mg·ha$^{-1}$ and year, but the average yield from the whole research period (2017–2019) amounted in average to ca. 13.5 Mg·ha$^{-1}$ and year and was lower due to habitat and weather conditions (drought in 2018 and in the first half year of 2019).

The method of plantation establishment, generative by sowing seeds (seeds) and vegetative by transplanting seedlings (planting), clearly influenced the obtained results. The DMY of plants established with the sowing method was higher (ca. 13.9 Mg·ha$^{-1}$·yr$^{-1}$) than the DMY of plants established with the planting method (ca. 13.0 Mg·ha$^{-1}$·yr$^{-1}$) due to higher number of shoots per unit area (1 m$^2$). Furthermore, the calorific value depended on the plantation establishment method. The biomass from plants established by the generative method had a higher calorific value (16.95 MJ·kg$^{-1}$) than biomass of plants established by the vegetative method (16.26 MJ·kg$^{-1}$).

The calorific value of cup plant biomass and other investigated parameters (moisture and ash content and content of macronutrients) achieved in this experiment does not differ from the calorific value and chemical properties of biomass other perennial species of energy crops. Especially the content of nitrogen, phosphorus, potassium and sulfur in *Silphium* biomass was low, and carbon content was high (34.0%), which makes the biomass of *Silphium* potentially a good quality raw material.

The total content of macronutrients in soil after the fourth year of *Silphium* cultivation was varied and generally smaller compared to the content from before establishing the experiment. However, *Silphium* cultivation has caused an increase in the content of available forms of phosphorus and potassium in the soil compared to content before establishing the experiment.

The results of our research indicated that to obtain a well-developed plantation with higher *Silphium* biomass yields of good quality, a better and cheaper method of plantation establishment is the generative method of sowing high-quality seeds, compared to the more complicated and cost-intensive vegetative method of planting.

**Author Contributions:** Conceptualization, M.B., E.M., T.K., H.S. and M.W.; Methodology, M.B., E.M., T.K., H.S. and M.W.; software, M.B.; validation, E.M., M.B. and T.K.; formal analysis, E.M., T.K., M.B., H.S.; investigation, M.B., E.M., T.K., H.S. and M.W.; resources, E.M., M.B., T.K.; data curation, M.B. and M.W.; writing preparation, E.M., M.B.; writing—original draft, M.B., E.M., T.K., H.S. and M.W.; review and editing, M.B.; visualization, M.B., E.M., T.K.; supervision, M.B. and T.K.; project administration, M.B.; funding acquisition, M.B. All authors have read and agreed to the published version of the manuscript.

**Funding:** This research (project "Novel Pathways of Biomass Production: Assessing the Potential of Sida hermaphrodita and Valuable Timber Trees", acronym SidaTim) received funding from the ERA NET Co-Fund FACCE SURPLUS under European Union's Horizon 2020 Research and Innovation Programme under grant agreement No 652615. and the National Centre for Research and Development (NCBR) in Warsaw (Poland) project No FACCE SURPLUS/I/SidaTim/03/2016.

**Acknowledgments:** We are very grateful to the European Union and to NCBR for the support that they have provided in allowing us to conduct this research. We thank all SidaTim project partners from Germany, UK, Italy, and especially the project coordinator, Michael Nahm. We are also thankful to the staff of the Agricultural Experiment Station Lipnik of the West Pomeranian University of Technology Szczecin for providing technical support for the field trials, and to Waldemar Piramowicz, Adam Sammel, Olga Kordula for their assistance with field work and Magdalena Sobolewska, Agata Skrobek and Magdalena Siwoń for the administrative support. We will thank the Polish and foreign students, especially Sheriff Noi from Ghana and Zeynep Kar from Turkey to help us by plants measurements.

**Conflicts of Interest:** The authors declare no conflict of interest. The funders had no role in the design of the study; in the collection, analyses, interpretation of data, in the writing of the manuscript or in the decision to publish the results.

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
