# Peer review of "Yields, Calorific Value and Chemical Properties of Cup Plant Silphium perfoliatum L. Biomass, Depending on the Method of Establishing the Plantation"

_agronomy, doi:10.3390/agronomy10060851_

Round 1

Reviewer 1 Report

All in all an interesting manuscript, which describes the cultivation under the given Polish conditions. Unfortunately, the origin of the plants is still somewhat diffuse and poorly defined. And it is not clear whether potted seedlings / plantlets or cuttings were planted. The differences in the results of sowing and planting could be discussed even more deeply.

The following things definitely need to be improved:

56–60 Why is only a part of the species mentioned? Many species such as S. terebinthinaceum, S. glutinosum, S. gracile, S. albiflorum, S. wasiotense and other species are missing. Some species  are misspelled: Silphium laciniatum du Silphium asteriscus are the correct spellings. Here should be a reference to the paper by Clevinger (JA Clevinger, JL Panero, Phylogenetic analysis of Silphium and subtribe Engelmanniinae (Asteraceae: Heliantheae) based on ITS and ETS sequence data, Am. J. Bot. 87 (2000) 565-572. ) appropriate. Or just give the number of species within the genus. S.perfoliatum is also found in Alabama, Kansas and North Dakota. In contrast, the species is not native in the eastern states and the northeast.“ Great Plains Area“ would be more appropriate than north-east part of USA.

74 green biomass

85, 172 and 548 contradiction cuttings or plantlets (=potted seedlings)?

109 air dryed

125 What's Polish classification? More specifically, please quote.

171–179 If the plant material has been propagated vegetativly , is it from selected plants? Do the differences in the results come from genetic differences? Or is both the same material? This is not clear here.

185 units are missing.

217-218 there is no reference to Andreas Schäfer's work

263 However,

293 and 309 Why two separate tables?

309 The number

311–327 How high was the point measurement? Why is the data compared with Wever's data 10 cm above the ground and not with 130 cm? This data from Wever (SD1 / 2) was collected in Rheinbach not in Dornburg.

324 However,

351–361 Does it make sense to show the SPAD data? You hardly have a statement.

404–414 Can this also be due to different genotypes (if cuttings(?) and seeds are not the same origin)?

234 Miscanthus sinensis not chinensis

471–472 plant biomass from planting

513–535 Where do the differences come from, discussion about rooting?

Author Response

Dear Reviewer,

thank you very much for your suggestions and comments. We added all appropriate changes and corrections and supplemented the article with relevant literature. 

Reviewer 2 Report

The paper is valuable in the context of energy plants exploration. It is comprehensive study on the quantitative and qualitative parameters of cup plant (Silphium perfoliatum), depending on the method of establishing the plantation. It presents also heating value and chemical composition of Silphium biomass as well as changes in soil parameters.

The Introduction and Methods sections are proper. The results are presented in analytical way, year by year, sometimes with too much details. The figures of Silphium in different years of development are very valuable, but it will be good to present the comparison between plants under two tested methods of establishing the plantation. The statistical analyses are proper in general (with comment see Line 234-235).

The Conclusions contain the most important results.

In References position no 36 is not cited in the text.

Linguistic correction of the paper is needed.

Other minor remarks are presented in the pdf of the paper.

Author Response

We thank very much the reviewer for the comments and for helping us to improve the manuscript and especially for some detailed remarks in the text. There were some editorial errors in the manuscript - we have all of them corrected.

In the statistical analyses we corrected also abbreviation of Tukey’s test – it is very helpful information about using properly names.

We checked the literature in manuscript and compliance with the References list and we changed also in References and in the text position 36.

We made some linguistic corrections.

Lines 234-235

Response: Done.

All remarks in the text in pdf were changed - done

We add also a statement to comparison between plants under two tested methods of plantation establishing. 

Thank you very much 

Marek Bury

Reviewer 3 Report

The cup plant Silphium perfoliatum L. is a plant originating in South America and Canada, therefore  in my opinion Authors should provide information on the origin of the seed.

Silphium perfoliatum L. is a perennial plant from the family Asteraceae and genus Silphium, which also includes Silphium trifoliatum L., Silphium integrifolium Michx. and Silphium lacinatum L., Silphium asteriseus L., Silphium radula L. Therefore, it would be correct to use Silphium perfoliatum L. (the Latin name) or cup plant (the English name). In addition, Latin names should be writtenin italics.

line 3. In my opinion the authors could not give brackets Silphium perfoliatum L.

line 240. Should be a paragraph

lines 115, 133, 135, 136, 160,161, 196…- should be °C, °F instead of  0C or 0F,

lines 233, 234…. Latin names should be written in italics.

Author Response

Dear Reviewer,

We thank you very much for the comments and for helping us to improve the manuscript.

We agree with your opinion and we made some changes in the text. 

R. The cup plant Silphium perfoliatum L. is a plant originating in South America and Canada, therefore  in my opinion Authors should provide information on the origin of the seed. 

Response: We wrote about origin of seed in Methods: the seeds are from Chrestensen Erfurter Samen- und Pflanzenzucht GmbH, Germany - line 174

Silphium perfoliatum L. is a perennial plant from the family Asteraceae and genus Silphium, which also includes Silphium trifoliatum L., Silphium integrifolium Michx. and Silphium lacinatum L., Silphium asteriseus L., Silphium radula L. Therefore, it would be correct to use Silphium perfoliatum L. (the Latin name) or cup plant (the English name). In addition, Latin names should be written in italics.

Response: Done. We agree with the reviewer and add information about genus. We know, that all botanical names should be written in Italics, but it was some oversight - we changed all Latin name in Italics

line 3. In my opinion the authors could not give brackets Silphium perfoliatum L. - 

Response: Done.

line 240. Should be a paragraph

Response: Done.

lines 115, 133, 135, 136, 160,161, 196…- should be °C, °F instead of  0C or 0F,

Response: Done.

lines 233, 234…. Latin names should be written in italics.

Response: Done.

Thank you very much and stay healthy

Marek Bury